# Elastically driven intermittent microscopic dynamics in soft solids

Mehdi Bouzid[1], Jader Colombo[1], Lucas Vieira Barbosa[1,2] & Emanuela Del Gado[1]

Soft solids with tunable mechanical response are at the core of new material technologies, but a crucial limit for applications is their progressive aging over time, which dramatically affects their functionalities. The generally accepted paradigm is that such aging is gradual and its origin is in slower than exponential microscopic dynamics, akin to the ones in supercooled liquids or glasses. Nevertheless, time- and space-resolved measurements have provided contrasting evidence: dynamics faster than exponential, intermittency and abrupt structural changes. Here we use 3D computer simulations of a microscopic model to reveal that the timescales governing stress relaxation, respectively, through thermal fluctuations and elastic recovery are key for the aging dynamics. When thermal fluctuations are too weak, stress heterogeneities frozen-in upon solidification can still partially relax through elastically driven fluctuations. Such fluctuations are intermittent, because of strong correlations that persist over the timescale of experiments or simulations, leading to faster than exponential dynamics.

[1] Department of Physics, Institute for Soft Matter Synthesis and Metrology, Georgetown University, 37th and O Streets, N.W., Washington District Of Columbia 20057, USA. [2] CAPES Foundation, Ministry of Education of Brazil, Brasilia - DF 70.040-020, Brazil. Correspondence and requests for materials should be addressed to E.D.G. (email: ed610@georgetown.edu).

Soft condensed matter (proteins, colloids or polymers) easily self-assembles into gels or amorphous soft solids with diverse structure and mechanics[1–5]. The nanoscale size of particles or aggregates makes these solids sensitive to thermal fluctuations, with spontaneous and thermally activated processes leading to rich microscopic dynamics. Besides affecting the time evolution, or aging, of the material properties at rest, such dynamics interplay with an imposed deformation and are hence crucial also for the mechanical response[6–10]. In the material at rest, thermal fluctuations can trigger ruptures of parts of the microscopic structure that are under tension or can promote coarsening and compaction of initially loosely packed domains, depending on the conditions under which the material initially solidified. The diffusion of particles or aggregates following such local reorganizations (often referred to as 'micro-collapses') is governed by a wide distribution of relaxation times, due to the disordered tortuous environment in which it takes place, that is, the microstructure of a soft solid, be it a thin fractal gel or a densely packed emulsion[11,12]. Therefore, localization, caging and larger-scale cooperative rearrangements in such complex environment will lead to slow, subdiffusive dynamics, similar to the microscopic dynamics close to a glass transition[13,14].

Several quasi-elastic scattering experiments or numerical simulations that can access micro- and nanoscale rearrangements in soft matter have indeed confirmed that the dynamics in aging soft solids are slower than exponential. Measuring the time decay of the correlations of the density fluctuations or of the local displacements of particles or aggregates, the experiments find that time correlations follow a stretched exponential decay $e^{-(t/\tau)^{\beta}}$ with $\beta < 1$, akin to the slow cooperative dynamics of supercooled liquids and glasses, in a wide range of gels and other soft solids[15–19]. Nevertheless, in the past few years there has been emerging evidence, through a growing body of similar experiments, that microscopic dynamics in the aging of soft materials can be instead faster than exponential. In a wide range of soft amorphous solids including colloidal gels, biopolymer networks, foams and densely packed microgels, time correlations measured via quasi-elastic scattering or other time-resolved spectroscopy techniques appear to decay as $e^{-(t/\tau)^{\beta}}$ with $\beta > 1$ (and in most cases $1.3 \leq \beta \leq 1.5$)[20–30]. When analysing the dependence of the relaxation time on the lengthscales probed, the data highlight the presence of super-diffusive rather than subdiffusive microscopic processes, with the relaxation time $\tau$ increasing with decreasing the scattering wave vector $q$ (and hence increasing the lengthscale being probed) as $\tau \propto 1/q$ rather than $\tau \propto 1/q^2$. Moreover, experiments specifically designed to quantify the fluctuations in time and space of the microscopic dynamics have revealed strong intermittency and pointed to the possibility of an abrupt and coherent reorganization of relatively large domains. Hence it has been suggested that the elastic rebound of parts of the material, after micro-collapses occur in its structure, should be, instead, the stress relaxation mechanism underlying the aging of soft solids[31–33]. As a matter of fact, viewing the soft glassy material as an elastic continuum and considering that the micro-collapses occur randomly in space (both strong assumptions in most of the materials of interest), the related disruption of the elastic strain field does lead to a compressed exponential relaxation regime, in relatively good agreement with experiments[33–36]. Understanding (and controlling) the nature of dynamical processes in aging soft solids is crucial to a wide range of technologies, from those involved in material processing to those relying on stable material properties. Moreover, the contrasting experimental findings also raise the very fundamental question of which microscopic mechanism should be prevalent in which material and under which conditions.

Here we use numerical simulations and a model particle gel which, during aging, undergoes structural changes by means of rupture of particle connections. Such ruptures are prototypical of the micro-collapses in the aging of such materials, occurring in regions where higher than average tensile stresses have been frozen-in during the solidification. By quantifying the consequences of the microscopic ruptures, we unravel that the origin of the different relaxation dynamics is in the dramatically different strain transmission if the stress heterogeneities are significantly larger than Brownian stresses. In those conditions, the characteristic timescale for stress relaxation through thermal fluctuations grows well beyond the duration of typical experiments or simulations, but the stress released in microscopic ruptures can still partially relax through elastically induced fluctuations, that are strongly correlated over time and through the structure. The emergent displacement distribution becomes scale-free, featuring a power law tail that points to the long-range elastic strain field in the gel as the main source of structural rearrangements. Thermal fluctuations and Brownian motion disrupt instead the spatial distributions of the local stresses, as well as the persistence of their correlations in time, which leads to the screening of the strain transmission over the structure. Our results provide a unifying framework to rationalize microscopic dynamical processes in a wide variety of soft solids, crucial to their mechanical performance.

## Results

**Model.** We consider a minimal 3D model for a gel composed of $N = 62{,}500$ colloidal particles of diameter $\sigma$ and mass $m$, with their motion described by a Langevin dynamics in the overdamped limit, at low volume fractions ($\simeq 7\%$) (see 'Methods' section). The particles spontaneously self-assemble into a disordered interconnected network thanks to a short range attractive well $U_2$ (combined with a repulsive core) and an additional short range repulsion $U_3$, which depends on the angle between bonds departing from the same particle and imparts an angular rigidity to the gel branches, as expected in such open structures[37–40]. Starting from the same gel configuration, we analyse the dynamics relevant to highly cohesive gels (where significant frozen-in stresses can be developed during solidification), by using different temperatures at fixed interaction strength $\epsilon$, to vary the ratio $k_B T/\epsilon$ of the Brownian stresses to stress heterogeneities frozen-in during solidification.

In spite of its simplicity, our model captures important physical features of real particle gels and can be used as a prototypical soft amorphous solid. The structural heterogeneities developed during the gel formation entail mechanical inhomogeneities and the simulations reveal the coexistence of stiffer regions (where tensile stresses tend to accumulate) with softer domains, where most of the relaxation occurs[41]. Hence aging is likely to occur via sudden ruptures of particle connections in the regions where the local tensile stresses are higher, as indeed suggested for the ultraslow aging observed in experiments[21]. In the simulations, they are quite rare even at the highest temperatures of interest here (roughly only a few over $\simeq 10^6$ MD steps at $k_B T/\epsilon = 0.05$ (ref. 39)) and become hardly observable on a reasonable simulation time window as we consider even smaller $k_B T/\epsilon$. To investigate the consequences of the ruptures on a timescale computationally affordable, we periodically scan the local stresses in the gel structure and remove particle bonds where the local stress is the highest, with a fixed rate $\Gamma$ (see 'Methods' section). Recombinations of the gel branches are possible but just not observed at the volume fractions and for the time window explored here.

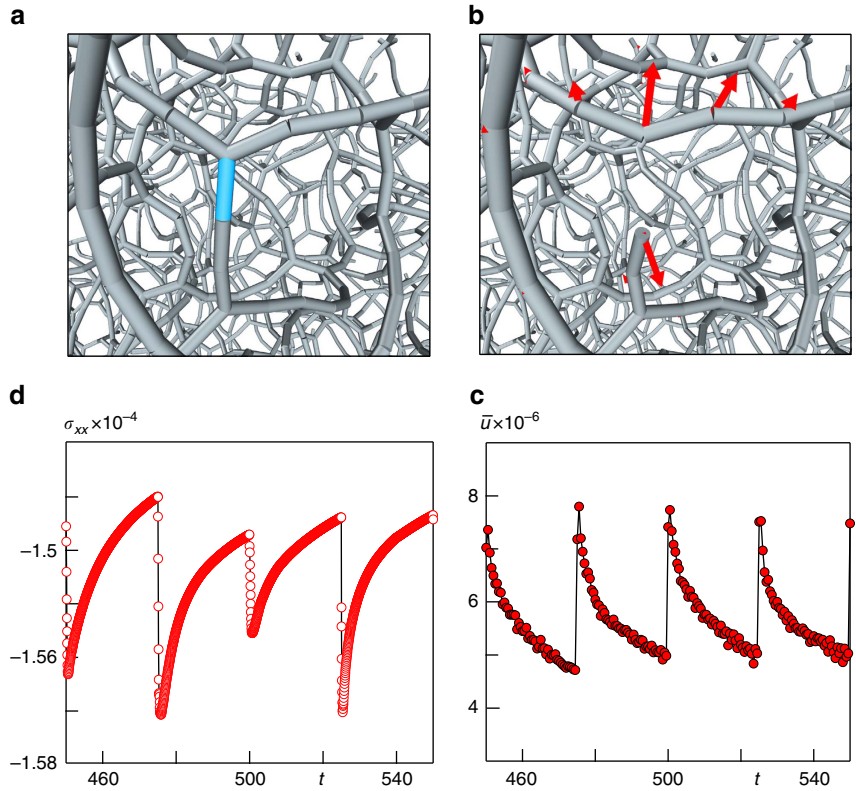

**Figure 1 | Rupture of gel connections.** Snapshots of the colloidal gel network showing the interparticle bonds before (**a**) and after (**b**) a bond rupture, each bond is represented by a segment, when the distance $d_{ij}$ between two particles $i$ and $j$ is $\leq 1.3\sigma$. In **a**, the bond that will break is highlighted, in **b**, the arrows indicate the displacement after the rupture. (**c**) The average displacement $\bar{u}$ as a function of the time for four consecutive ruptures. (**d**) The corresponding stress component $\sigma_{xx}$ as a function of the time.

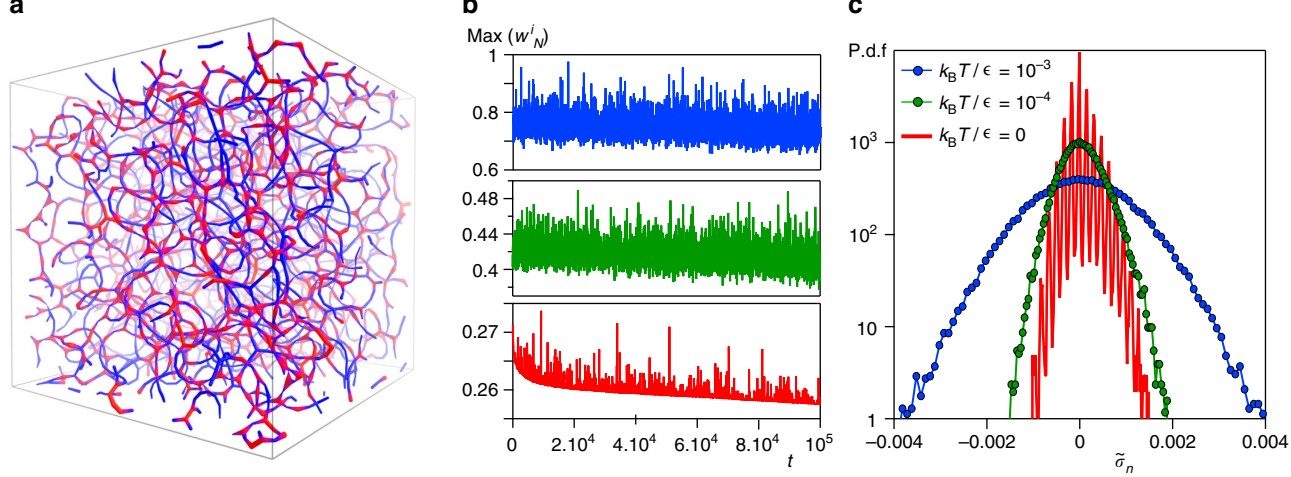

**Figure 2 | Local stresses.** (**a**) A snapshot of the initial colloidal gel network for $k_B T/\epsilon = 0$ and at low volume fraction $\phi = 7\%$, showing the interparticle bonds represented by a segment when the distance $d_{ij}$ between two particles $i$ and $j$ is less than $1.3\sigma$. The colours indicate the local normal stress $\tilde{\sigma}_n$, using red for tension and blue for compression. (**b**) Time evolution of the maximum contribution (per particle) $w_N^i$ to the normal component of the stresses tensor for the athermal network (red), $k_B T/\epsilon = 10^{-4}$ (green) and $k_B T/\epsilon = 10^{-3}$ (blue). (**c**) The final probability distribution function of $\tilde{\sigma}_n$ corresponding to the initial configuration shown in **a**. P.d.f, probability distribution function.

In a rupture event, illustrated in Fig. 1a,b using two subsequent simulation snapshots, the particles depart from each other and move following the part of the network to which they are attached. The topological and elastic constraints exerted by the network, coupled to the local dissipation, result in a progressive slowing down of such motion. The average displacement $\bar{u} = 1/N \sum_{i=1}^{N} \| r_i(t+\delta t) - r_i(t) \|$ ($\delta t$ is the integration timestep in the simulations) decreases exponentially between two rupture events, as shown in Fig. 1c for $k_B T/\epsilon = 0$. After each event, interparticle forces and/or thermal fluctuations drive the redistribution of the stresses over the structure, with the stress relaxation shown in Fig. 1d using the component $\sigma_{xx}$ of the virial

stress tensor (that is, the stress computed from the microscopic particle configurations[42], see 'Methods' section) plotted as a function of the time. All data shown in the following refer to a removal (or cutting) rate $\Gamma = 0.04\tau_0^{-1}$, where $\tau_0 = \sqrt{m\sigma^2/\epsilon}$ is the MD time unit, and correspond to breaking $\simeq 5\%$ of the initial bonds in total over the simulation time window of $10^5\tau_0$ considered here. We have carefully investigated how the choice of $\Gamma$ affects the results discussed below and determined that they do not significantly change as long as the removal rate is high enough to collect enough statistics (of the ruptures and their consequences) but slow enough to allow for a partial stress relaxation between two rupture events (Fig. 1d) (see 'Methods' section).

**Internal stresses.** The analysis of the stress heterogeneities in the network helps disentangle how the stress reorganization and the presence of thermal fluctuations contribute to the relaxation dynamics. Because of the topological constraints produced in the network on its formation, in the initial configuration local stresses have significant spatial correlations. Particles located in the nodes of the gel network, in fact, contribute to the local stresses mostly in terms of tensions, as shown in Fig. 2a, where the gel is coloured according to the value of the local normal stress $\tilde{\sigma}_n$ (that is, the normal virial stress computed over a small volume around each particle and containing $\sim 10$ particles, see 'Methods' section). Analyzing the evolution over time of the largest among the contributions $\omega_n^i$ of each particle $i$ to the total normal stress $\sigma_n$, we find that it progressively decreases in all cases, but its fluctuations are clearly different in nature depending on the relative strength of thermal fluctuations (see Fig. 2b). The decrease rate of $\omega_n^i$ clearly depends on $k_BT/\epsilon$ and is much slower than the cutting rate, indicating that the aging of the material corresponds to a slower, large-scale stress redistribution, which emerges over much longer timescales. In the athermal case, this slow evolution of the structure is found to be logarithmic in time. The final distribution of local stresses $\tilde{\sigma}_n$ (Fig. 2c) still features a few of the pronounced peaks present in the initial one, pointing to the persistence, during the aging, of the spatial correlations of the local stresses. Thermal fluctuations, instead, clearly disrupt such correlations and help redistribute the local stresses much more homogeneously over the gel structure.

**Microscopic dynamics.** The mechanical heterogeneities and their time evolution during the aging have an impact on the relaxation dynamics. To quantify it, we use the same observables measured in quasi-elastic scattering experiments. From the particle coordinates, we compute the coherent scattering function:

$$F_{t_w}(q,\tau) = \frac{1}{NS(q)} \sum_{j,k=1}^{N} \exp\left[-i\mathbf{q} \cdot \left(\mathbf{r}_k(\tau+t_w) - \mathbf{r}_j(t_w)\right)\right],$$

where $S(q)$ is the structure factor of the gel and $t_w$ the time at which the measurement starts (here $t_w = 0$ for all data sets). From the long time decay of $F(q,\tau)$ ($= F_{t_w=0}(q,\tau)$), we extract the relaxation time $\tau(q)$ for different wave vectors (see 'Methods' section) and in Fig. 3a we plot $F(q,\tau)$ as a function of $\tau/\tau(q)$, for $q$ varying between 0.1 and 10 (in units of $\sigma^{-1}$), at three different temperatures: $k_BT/\epsilon = 10^{-3}$ (blue), $k_BT/\epsilon = 10^{-4}$ (green) and for the athermal network (red). The data indicate that the form of the time decay is sensitive not only to $q$ (as a consequence of the structure heterogeneities)[31,43,44], but also to $k_BT/\epsilon$. In all cases, the final part of $F(q,\tau)$ is well fitted by a decay $\propto \exp\left[-(\tau/\tau(q))^\beta\right]$. The exponent $\beta$, extracted from the best fit of the past decade of $F(q,\tau)$, indicate a transition from a stretched to a compressed exponential decay upon decreasing the

ratio $k_BT/\epsilon$ (see Fig. 3b). The dependence of $\beta$ on $k_BT/\epsilon$ found here is consistent with experimental observations in refs 25,26. The dependence on $q$ is the same as typically observed in several experiments[22,28,31], with the $\beta$ decreasing rapidly with increasing $q$ over lengthscales typically ranging between a few and several particle diameters, up to roughly the order of the mesh size of the gel network (the mean distance between two network nodes along a gel branch is $2\pi/q \simeq 5\sigma$ at this volume fraction[39]). For much larger distances, $\beta$ does not change much with $q$. The particle mean squared displacement (MSD) is shown in Fig. 3c as a function of the time for the different values of $k_BT/\epsilon$. After a localization process at intermediate times, the MSD displays a change in the time dependence at long times, from subdiffusive to super-diffusive, on decreasing $k_BT/\epsilon$. The wave vector dependence of the relaxation time $\tau(q)$ is reported, for different $k_BT/\epsilon$, in Fig. 3d. For $q$ corresponding to distances up to $\simeq 5\sigma$, $\tau(q)$ is strongly dependent on $q$ and on $k_BT/\epsilon$ and, as long as $k_BT/\epsilon \simeq 10^{-3}$, it follows the scaling $\propto 1/q^2$ typically expected when diffusive processes are at play. Reducing $k_BT/\epsilon$ leads to a different scaling $\tau(q) \propto 1/q$, which is consistent with the super-diffusive microscopic motion detected in Fig. 3c. The results obtained for $k_BT/\epsilon < 10^{-3}$ are consistent with several experiments, finding a compressed exponential dynamics and a $1/q$ scaling of the relaxation time. Such dependence was ascribed to a super-diffusive, quasi-ballistic particle motion in refs 21,31,45, a hypothesis not directly testable in experiments but proven in Fig. 3c. The ballistic motion reported here and in experiments persists over timescales corresponding to a large number of events. This feature points to the fact that the origin of the ballistic motion is not the single breaking event but the elastic relaxation accumulated in the network over many events.

The agreement between our scenario for small enough $k_BT/\epsilon$ and the experiments support the idea that frozen-in stresses control the microscopic dynamics also in the experimental samples[23,46]. The relaxation time $\tau(q)$ from the simulations becomes less sensitive to the wave vector over distances beyond the mesh size of the network, as indeed observed in experiments[20] (see Fig. 3d). Whereas such finding could be interpreted as a loss of dynamical correlations over large distances, we have verified through mechanical tests that the gels are prevalently elastic for all values of $k_BT/\epsilon$ considered here, supporting the presence of long-range spatial correlations of the dynamics. Moreover, the relaxation time extracted from the incoherent scattering keeps instead increasing on decreasing $q$ and also indicate long-range spatial correlations in the dynamics (see Supplementary Fig. 1). Hence the data in Fig. 3d can be better understood by considering that, although local rearrangements are possible, over larger distances the material is solid and elastically connected in spite of being structurally heterogeneous and sparse, and the time correlations of the density fluctuations over those lengthscales are therefore locked-in.

Beyond the capability of experiments, the simulations give us direct access to the distribution of the displacements over a lag time in between two rupture events, which, for $k_BT/\epsilon > 10^{-3}$, is a bell-shaped curve with exponential tails (Fig. 4a), similar to those typically seen in supercooled liquids[38,47,48]. However, on decreasing $k_BT/\epsilon \to 0$, the probability distribution function becomes instead progressively scale-free. Such findings support our interpretation of Fig. 3d and the idea that the dynamics are spatially correlated and cooperative over extended domains for all $k_BT/\epsilon$. We visualize the displacement field in two representative snapshots of the gel network (Fig. 4b,c) where the colour code indicates the amplitude of the displacements measured along the network after a rupture event, for $k_BT/\epsilon = 10^{-3}$ and $k_BT/\epsilon = 0$, respectively. The visualizations highlight how the long-range elastic response of the network dominates the

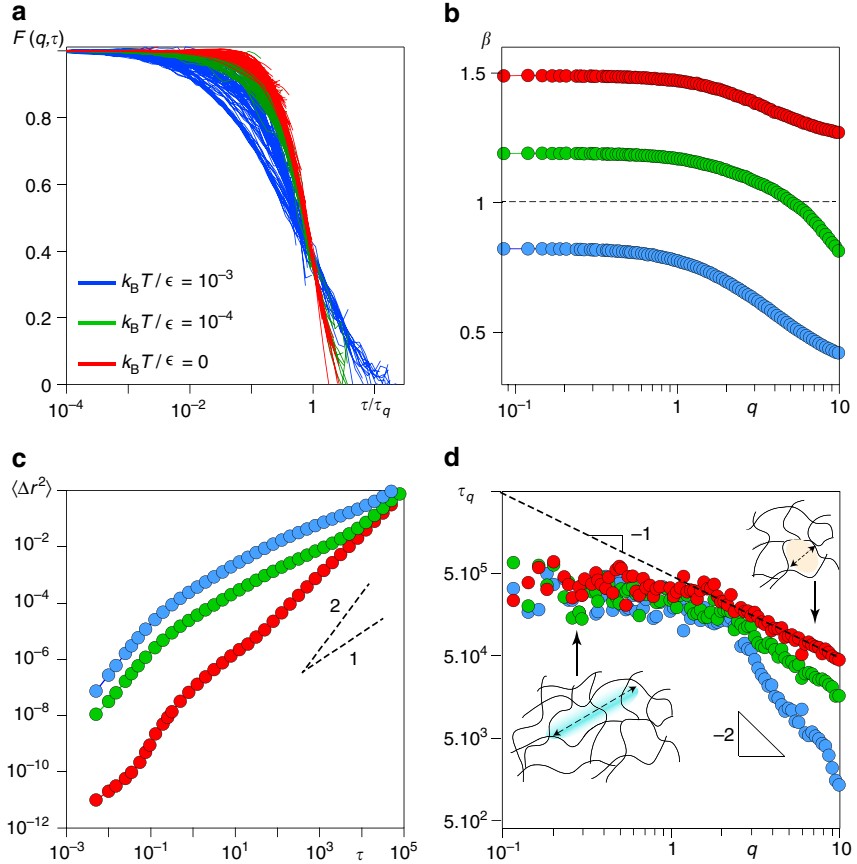

**Figure 3 | Relaxation dynamics.** (**a**) The decay of the coherent scattering function as a function of the time rescaled by the relaxation time $\tau_q$ for a wave vector ranging from $q = 0.1$ to $10$ and for three different ratios of $k_B T/\epsilon$. (**b**) The exponent $\beta$ as a function of the wave vector $q$, showing a transition from a compressed to a stretched dynamics. (**c**) The particle MSD as a function of time $t$ for the same $k_B T/\epsilon$ ratios as in **a**. (**d**) The relaxation time $\tau(q)$ as a function of the wave vector $q$ for the three systems: the scaling goes from $\tau \sim q^{-1}$ for the fully athermal regime to diffusive $\tau \sim q^{-2}$ when the thermal fluctuations dominate; a plateau emerges at low wave vectors corresponding to distances beyond the mesh size of the network, as sketched in the cartoons.

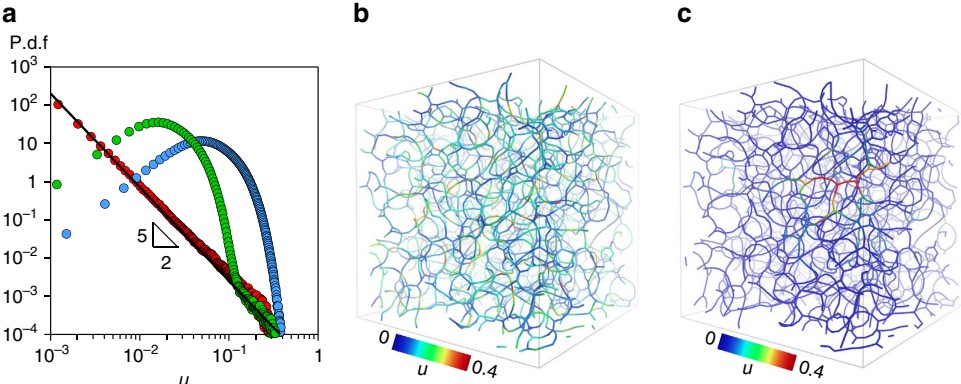

**Figure 4 | Microscopic particle displacements.** (**a**) The probability distribution function of the particle displacements calculated with a lag time equal to $40\tau_O$ for three ratios of $k_B T/\epsilon$, blue circles represent the fully thermal regime $k_B T/\epsilon = 10^{-3}$, green circles refer to the intermediate regime $k_B T/\epsilon = 10^{-4}$ and red circles correspond to the configuration at $k_B T/\epsilon = 0$, the solid black line represents the MF prediction $u^{-5/2}$, a deviation from this purely elastic effects is seen when thermal fluctuations act on the small displacements. Snapshots of the colloidal gel network for $k_B T/\epsilon = 10^{-3}$ (**b**) and $k_B T/\epsilon = 0$ (**c**), showing the interparticle bonds and the amplitude of the displacements after a rupture event.

displacement pattern in the athermal limit, where it extends indeed over several network meshes, and it is instead screened by the small displacements induced by thermal fluctuations for larger $k_B T/\epsilon$ (see Supplementary Movies).

For a homogeneous elastic material, the amplitude of the strain $u$ induced at a distance $r$ from the rupture event decreases as $u(r) \propto 1/r^2$ (ref. 50). Given the probability $P(r)$ of having a rupture event at a distance $r$ and if, in a mean field (MF) approximation, rupture events occur anywhere with the same probability, the probability distribution function for a displacement (or strain) of amplitude $u$ can be estimated as $P(u) \sim P(r) \mid dr/du \mid \sim r^2 r^3 \sim u^{-5/2}$. In the athermal limit

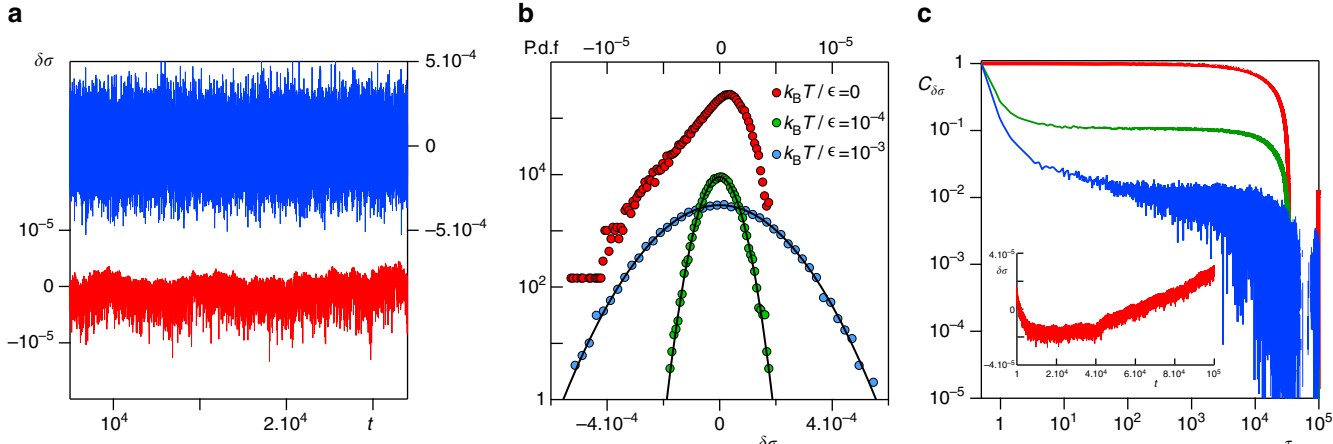

**Figure 5 | Stress fluctuations. (a)** Time series of the normal stress fluctuations $\delta\sigma = \sigma_{xx} - \langle\sigma_{xx}\rangle$ measured over a time interval involving 800 ruptures for the athermal network (red) and for $k_BT/\epsilon = 10^{-3}$ (blue). The corresponding probability distribution function **(b)**: in the regime dominated by frozen-in stresses (red), the stress fluctuations are elastically driven and intermittent in nature. **(c)** Main frame: The stress fluctuations autocorrelation function measured over the whole simulation for the fully thermal regime $k_BT/\epsilon = 10^{-3}$ (blue), the intermediate $k_BT/\epsilon = 10^{-4}$ (green) and the for $k_BT/\epsilon = 0$ (red). Inset: Time series of the normal stress fluctuations $\delta\sigma$ over all the simulation for the athermal sample showing the aging of the structure. P.d.f, probability distribution function.

$k_BT/\epsilon = 0$, we clearly recover the MF prediction with the power law $u^{-5/2}$ (Fig. 4a), which is in fact compatible with a compressed exponential decay of the scattering functions with $\beta = 3/2$ (refs 33,35), as shown in Fig. 3b. The agreement with the MF prediction may suggest that structural disorder and correlations between rupture events are in the end negligible. Nevertheless, here (as well as in many experiments) the structure is heterogeneous, the correlations of events are crucial and the MF assumptions are unlikely to truly hold[35]. We propose therefore a different explanation, that is, structural disorder and correlations between rupture events, in spite of being present, do not disrupt the elastic connectivity of the material because, when $k_BT/\epsilon \sim 0$, the relaxation of the stresses through elastic fluctuations does not weaken their spatio-temporal correlations. As a matter of fact, in addition to the spatial correlations discussed above, the total stress also features large fluctuations in time, that depend on $k_BT/\epsilon$ and become strongly correlated as $k_BT/\epsilon \to 0$ (Fig. 5).

**Stress fluctuations and stress relaxation.** The fluctuations of one component of the total stress ($\delta\sigma = \sigma_{xx} - \langle\sigma_{xx}\rangle$), measured over a time window significantly larger than the time interval between two rupture events, are plotted in Fig. 5a for $k_BT/\epsilon = 10^{-3}$ and $k_BT/\epsilon = 0$. When Brownian stresses become negligible with respect to the stress heterogeneities, such fluctuations clearly deviates from randomly fluctuating values, even after many ruptures have occurred. Their probability distribution function is Gaussian for larger $k_BT/\epsilon$, whereas it develops non-Gaussian tails as $k_BT/\epsilon \to 0$ (Fig. 5b). The non-Gaussian processes and the intermittent fluctuations measured over the same time window correspond indeed to persistent time correlations (see Supplementary Fig.2).

When monitoring the stress fluctuations over the whole duration of the simulations, we find evidence of the progressive aging of the structure that eventually emerges from the correlated stress redistribution (see Fig. 5c, inset for $k_BT/\epsilon = 0$ data). Accordingly, the time correlations of the fluctuations of the total stress over these longer timescales develop a pronounced plateau on decreasing $k_BT/\epsilon$ and eventually extend to the whole duration of the simulation in the athermal limit (Fig. 5c, main frame).

Time- and space-resolved scattering experiments have indeed found evidence of highly intermittent dynamical processes[43],

hence Fig. 5a,b and the intermittency in the stress fluctuations detected here provide a first coherent physical picture also for these observations, beyond the aspects that can be captured by MF approaches. Our analysis reveals that indeed the spatio-temporal correlation patterns detected in the microscopic displacements and in the density fluctuations (Figs 3a–c and 4a) stem from the way thermal fluctuations, on increasing $k_BT/\epsilon$, help redistribute elastic stresses (and strains) in the material. When thermal fluctuations are weak, the microscopic dynamics emerge instead from elastically driven stress fluctuations that remain strongly correlated in space and time.

## Discussion

Our analysis provides a vivid microscopic picture for the hypothesis underlying recent theories of aging in soft solids[34,35] and has implications for a potentially wider range of materials.

The control parameter $k_BT/\epsilon$ simply reflects the ratio between the timescales governing stress relaxation, respectively, through thermal fluctuations ($\eta\sigma^3/k_BT$) and elastic recovery ($\eta\sigma^3/\epsilon$) in the material ($\eta$ being the viscous damping). Hence it helps identify the conditions for which the elastically driven intermittent dynamics emerge in different jammed solids. First, we note that while we have considered here only one type of micro-collapses, the distruption of the elastic strain field due to the rupture of a branch of the gel is basically the same, in a first approximation, as the one induced by a recombination of the gel branches (or by other types of possible microscopic events that help relax frozen-in stresses)[33]. Therefore, we expect our general picture to have a much wider relevance. Second, the physical mechanisms we propose help rationalize several experimental observations in very different materials, ranging from biologically relevant soft solids to metallic glasses[31,45,46,50]. In a quasi-equilibrium scenario, enthalpic and thermal degrees of freedom may still couple and stress correlations decay relatively fast. When the material is deeply quenched and jammed, instead, recovering the coupling between the distinct degrees of freedom and restoring equilibrium will require timescales well beyond the ones accessible in typical experiments or simulations. The result will be intermittent dynamics and compressed exponential relaxations. The competition between Brownian motion and elastic effects through the relaxation of internal stresses illustrated

in this work suggests different scenarios for the energy landscape underlying the aging of soft jammed materials. When thermal fluctuations screen the long-range elastic strain transmission, microscopic rearrangements may open paths to deeper and deeper local minima in a rugged energy landscape. Compressed exponential dynamics, instead, evoke the presence of flat regions and huge barriers, with the possibility of intermittent dynamics, abrupt rearrangements and avalanches[34,35]. Investigating how such different dynamical processes couple with imposed deformations will provide a new rationale, and have important implications, for designing mechanics, rheology and material performances.

## Methods

**Numerical model and viscoelastic parameters.** The particles in the model gel interact through a potential composed of two terms. The first force contribution derives from a Lennard–Jones-like potential of the form:

$$\mathcal{U}_2(\mathbf{r}) = A\left(\frac{a}{r^{18}} - \frac{1}{r^{16}}\right). \tag{1}$$

The second contribution confers an angular rigidity to the inter particles bonds and takes the form:

$$\mathcal{U}_3(\mathbf{r}, \mathbf{r}') = B\Lambda(r)\Lambda(\mathbf{r}')\exp\left[-\left(\frac{\mathbf{r}\cdot\mathbf{r}'}{rr'} - \cos\theta\right)^2 w^{-2}\right]. \tag{2}$$

The strength of the interaction is controlled by $\Lambda(r)$ and vanishes over two particles diameters:

$$\Lambda(r) = r^{-10}\left[1 - (r/2)^{10}\right]^2 \Theta(2-r). \tag{3}$$

where $\Theta$ denotes the Heaviside function. The evolution of the gel over time is obtained by solving the following Langevin equation for each particle:

$$m\frac{d^2\mathbf{r}_i}{dt^2} = -\nabla_{r_i}\mathcal{U} - \eta_f\frac{d\mathbf{r}_i}{dt} + \xi(t), \tag{4}$$

$$\mathcal{U}(\mathbf{r}_i, ..., \mathbf{r}_N) = \epsilon\left[\sum_{i>j}\mathcal{U}_2(\frac{\mathbf{r}_{ij}}{\sigma}) + \sum_i\sum_{j>k}^{j,k\neq i}\mathcal{U}_3(\frac{\mathbf{r}_{ij}}{\sigma}, \frac{\mathbf{r}_{ik}}{\sigma})\right] \tag{5}$$

where $\sigma$ is the particle diameter, $\xi(t)$ is a random white noise that models the thermal fluctuations and is related to the friction coefficient $\eta_f$ by means of its variance $\langle\xi_i(t)\xi_j(t')\rangle = 2\eta_f k_B T\delta_{ij}\delta(t-t')$. To be in the overdamped limit of the dynamics $\eta_f$ is set to 10, and the timestep $\delta t$ used for the numerical integration is $\delta t = 0.005$. The parameters of the potential are chosen such that the disordered thin percolating network starts to self assemble at $k_B T/\epsilon = 0.05$. One convenient choice to achieve this configuration is given by this set of parameters: $A = 6.27$, $a = 0.85$, $B = 67.27$, $\theta = 65°$ and $w = 0.3$. The system is composed of $N = 62,500$ particles in a cubic simulation box of a size $L = 76.43\sigma$ with periodic boundary conditions, the number density $N/L^3$ is fixed at 0.14, which corresponds approximately to a volume fraction of 7%. All initial gel configurations are the same, prepared with the protocol described in ref. 40, which consists in starting from a gas configuration ($k_B T/\epsilon = 0.5$) and letting the gel self-assemble upon slow cooling down to $k_B T/\epsilon = 0.05$. We then quench this gel configuration by running a simulation with the dissipative dynamics $m\frac{d^2\mathbf{r}_i}{dt^2} = -\nabla_{\mathbf{r}_i}\mathcal{U} - \eta_f\frac{d\mathbf{r}_i}{dt}$ until the kinetic energy drops to zero($10^{-24}$). All simulations have been performed using a version of LAMMPS suitably modified by us[52].

**Stress calculation and cutting strategy.** We let the initial gel configuration evolve with equation (4) for each of the different values of $k_B T/\epsilon$ considered here, while using the following procedure to cut network connections. At each timestep, we characterize the state of stress of a gel configuration by computing the virial stresses as $\sigma_{\alpha\beta} = -\frac{1}{V}\sum_i w^i_{\alpha\beta}$, where the Greek subscripts stand for the Cartesian components $x$, $y$, $z$ and $w^i_{\alpha\beta}$ represents the contribution to the stress tensor of all the interactions involving the particle $i$ and $V$ is the total volume of the simulation box[52]. $w^i_{\alpha\beta}$ contains, for each particle, the contributions of the two-body and the three-body forces evenly distributed among the particles that participate in them:

$$w^i_{\alpha\beta} = -\frac{1}{2}\sum_{n=1}^{N_2}\left(r^i_\alpha F^i_\beta + r'_\alpha F'_\beta\right) + \frac{1}{3}\sum_{n=1}^{N_3}\left(r^i_\alpha F^i_\beta + r'_\alpha F'_\beta + r''_\alpha F''_\beta\right) \tag{6}$$

The first term on the RHS denotes the contribution of the two-body interaction, where the sum runs over all the $N_2$ pair of interactions that involve the particle $i$. ($r^i$, $F^i$) and ($r'$, $F'$) denote, respectively, the position and the forces on the two interacting particles. The second term indicates the three-body interactions involving the particle $i$. We consider a coarse-graining volume $\Omega_{cg}$ centered around the point of interest $r$ and containing around 9–10 particles on average, and define the local coarse-grained stress based on the per-particle virial contribution as

$\bar{\sigma}_{\alpha\beta}(\mathbf{r}) = -\sum_{i\in\Omega_{cg}} w^i_{\alpha\beta}/\Omega_{cg}$. For a typical starting configuration of the gel, the local normal stress $\bar{\sigma}_n = (\bar{\sigma}_{xx} + \bar{\sigma}_{yy} + \bar{\sigma}_{zz})/3$ reflect the heterogeneity of the structure and tend to be higher around the nodes, due to the topological frustration of the network. We consider that breaking of network connections underlying the aging of the gel is more prone to happen in the regions where local stresses tend to be higher, as found also in refs 40,41. Hence, to mimic the aging in the molecular dynamics simulations we scan the whole structure of the gel and remove one of the bonds (by turning off the well in $\mathcal{U}_2$) whose contribution to the local normal stress $w^i_n = \left(w^i_{xx} + w^i_{yy} + w^i_{zz}\right)/3$ is the largest (prevalently bonds between particles belonging to the network nodes). As the simulation proceeds, local internal stresses redistribute in the aging structure of the gel and the locations of more probable connection rupture (as well as their number) change over time. All simulations discussed here have been performed with a rate $\Gamma = 0.04\tau_0^{-1}$, corresponding to removing only ∼5% of the total network connections over the whole simulation time window. We have run several tests varying the removal rate in order to verify that changing the rate (having kept all other parameters constant) does not modify our outcomes and the emerging physical picture. Overall, varying $\Gamma$ over nearly two orders of magnitudes, we recover the same results, as long as the $\tau_r = 1/\Gamma$ between two rupture events allows for at least partial stress relaxation (see Fig. 1).

**Stress autocorrelation function.** The autocorrelation function of the stress fluctuations $\delta\sigma = \sigma_{xx} - \langle\sigma_{xx}\rangle$ is computed as follows: $C_{\delta\sigma} = f_{\delta\sigma}(\bar{\tau})/f_{\delta\sigma}(0)$, where $f_{\delta\sigma}$ is the auto-covariance and takes the form $f_{\delta\sigma}(\bar{\tau}) = \sum_{t=0}^{n-\bar{\tau}}(\sigma_{xx}(t+\bar{\tau}) - \langle\sigma_{xx}\rangle)(\sigma_{xx}(t) - \langle\sigma_{xx}\rangle)$ and $\langle\sigma_{xx}\rangle = \frac{1}{n+1}\sum_{t=0}^n\sigma_{xx}(t)$. We have computed the autocorrelation function of the stress fluctuations over partial time series and over the whole duration of the simulations (see Fig. 5 and Supplementary Fig. 2).

**Fitting procedure for the intermediate scattering functions.** To extract the $\beta$ exponent and the structural relaxation time $\tau_q$, we fit the last decay of the coherent scattering function $F(q,t)_{t_{w=0}}$ simultaneously with the incoherent scattering function $F_s(q,t)_{t_{w=0}}$, defined as $F_s(q,t) = \sum_{k=1}^N \exp[-i\mathbf{q}\cdot(\mathbf{r}_k(t) - \mathbf{r}_k(0))]/N$. This observable is less noisy and has the same exponent as the coherent scattering function shown in Fig. 3 (see also Supplementary Fig. 3). Having fitted $F_s(q,t)$ with a stretched exponential type of the form $A + B\exp(-t/\tau_s)^\beta$ and having extracted the exponent $\beta$, such exponent is used to initiate the fit of $F(q,t)$ by locking this parameter and letting all the others free. This procedure helps us to improve the quality of the fit used to extract the relaxation time $\tau_q$ from the coherent scattering.

**Data availability.** The authors declare that all data supporting the findings of this study are available within the article and its Supplementary Information Files. All raw data can be accessed on request to the authors.

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

## Acknowledgements

The authors thank Jean-Louis Barrat, Ludovic Berthier, Luca Cipelletti, Thibaut Divoux, Ezequiel Ferrero, Kirsten Martens, Peter D. Olmsted and Veronique Trappe for insightful discussions. L.V.B. was supported from CAPES Foundation, Ministry of Education of Brazil (Proc. N 88888.059093/2013-00). E.D.G. acknowledges support from the Swiss National Science Foundation (Grant No. PP00P2 150738), the National Science Foundation (Grant No. NSF PHY11-25915). All authors thank Georgetown University.

## Author contributions

M.B., J.C. and E.D.G. designed the research; M.B. performed the research and analysed the data; J.C. and M.B. developed analytical tools; L.V.B. developed visualization tools. M.B. and E.D.G. wrote the paper.

## Additional information

**Competing interests:** The authors declare no competing financial interests.

