## [Peer Review File · Nature Communications]

Reviewers' comments:

Reviewer #1 (Remarks to the Author):

In this work numerical simulations are carried out to model aging in a particulate gel as a rupture of interparticle bonds. Being an experimentalist, I have some genuine concerns on implications of this work.

1. Authors show that in an athermal limit system shows compressed exponential relaxation while at higher temperatures it transforms to stretched exponential. Are there any experimental reports that support this scenario, where with increase in temperature, β is observed to decrease (from >1 to <1) over the same q .

2. Authors cite many papers wherein compressed exponential behavior is observed (references: 18 to 25). However in most of these systems aging involves either increase in elastic modulus or that of relaxation time. While authors do plot relaxation time dependence of q , nowhere they plot relaxation time dependence of aging time. How does it look like ? Does it resemble scalings mentioned in any of the references 18-25.

3. My bigger concern is that the authors express aging as rupture of interparticle bonds. However, experimentally enhancement of relaxation time/elastic modulus is observed. Doesn't it mean that with time there should be net buildup of bonds than rupture. Authors should therefore plot evolution of relaxation time/elastic modulus as a function of time for the three cases explored.

Reviewer #2 (Remarks to the Author):

In this paper, the authors report on a large-scale study of a model gel through overdamped Langevin dynamics. They identify and isolate those links in the gel that carry the largest stress, and study what happens when these links are artificially removed. They find that a qualitatively different dynamics at high and low temperatures. At high temperature, the dynamics is subdiffusive with stretched exponentials, as in regular glassy materials. At low temperature, it becomes superdiffusive and shows compressed exponential decay, as in some experiments on gels that appeared in the literature recently. They attribute this difference to a change in the way elastic stresses propagate in the material. At high temperature, thermal fluctuations screen the elastic propagator leading to a Gaussian like distribution of stresses. At low temperature this screening becomes ineffective leading to a power-law distribution of stresses and a faster propagation. (This is at least my understanding of the paper).

I find the result very interesting. It provides a general picture to understand the difference between glasses and gels, based on a clear microscopic phenomenon. The study will probably have an important impact on the community and lead to many follow-ups. I therefore recommend publication in Nature Communications.

The results are clearly explained and the analysis looks solid and reproducible.

The introduction and conclusions to the paper could instead be improved. As such, they are quite difficult to read for a non specialist. Many technical words are used without definition (e.g. "enthalpic stresses", which by the way is a concept that is never fully defined and I did not understand the reason for this name). I would suggest a longer introduction, with a better explanation of the main concepts and a more explicit contrast between the standard glassy scenario and the new scenario proposed here.

Reviewer #3 (Remarks to the Author):

REFEREE REPORT on

"Elastically driven, intermittent microscopic dynamics in soft solids"

by M. Bouzid et al.

In the present manuscript (MS) the authors present a detailed numerical investigation of a specific model describing a low density gel with directional bonds. The authors investigate the dynamics during aging at different "temperature", to change the relative importance of thermal fluctuations and mechanical relaxation. Investigating the stress and the displacements associated to rupture events during the dynamics, the authors find the transition from a thermal fluctuation dominated aging at high T to a more mechanical (enthalpy) dominated aging in the athermal system. In the latter regime, the fluctuations are strongly non Gaussian. According to the authors this model well represents a class of real systems, allowing to rationalize the experimental findings. Moreover this model allows one to study the aging processes in soft matter, an issue of great importance for the practical application of these materials.

I find that the work is highly professional and the MS is well written (few minor issues are listed below). Overall, I think that the MS deserves to be published in Nature Communications,

There is, however, one issue that puzzle me. Concerning the intermediate scattering function, the authors find that at small q the relaxation time is q-independent (see Fig. (3), panel d). This is a strange result, not shared with other (simulated or experimental) cases (Ref. [18], cited by the authors at support of this finding, concerns fractal objects, cannot be considered a good counterpart for the present model). Intuitively the flatness of $\tau(q)$ at small q should correspond to a dynamics made of largely independent and uncorrelated (over large distance) events. How this is compatible with the authors' statement: "Such finding suggests that, although local rearrangements are possible, over larger distances the material is solid and elastically connected in spite of being structurally heterogeneous and sparse...". My feeling is that the investigated model has a dynamic basically uncorrelated over large distances. I would ask the authors to further expand this point. Indeed, the interest on the present results (i.e. its applicability to the science of materials) strongly depends on how much its dynamic is far from a collection of (spatially) isolated events.

Minor issues:

1-In the displayed equation (section "Microscopic dynamics") defining $F_{\{t_w\}}(q, \tau)$, in the argument of the exponential appears $(r_k(\tau) - r_j(t_w))$. Should not it be $(r_k(t_w + \tau) - r_j(t_w))$?

2-The labelling (a, b, c) of the three panels of Figure 4 is wrongly reported in the caption.

3-Figure 5. I suppose that the color code for the three panels is the same reported in the central one. For clarity, please, report the code in the caption..

Reply to referee 1:

Question : Authors show that in an athermal limit system shows compressed exponential relaxation while at higher temperatures it transforms to stretched exponential. Are there any experimental reports that support this scenario, where with increase in temperature, β is observed to decrease (from > 1 to < 1) over the same q .

ANSWER: The control parameter in our study is in fact $k_B T/\epsilon$, i.e. the ratio of thermal fluctuations to interaction strength, rather than just the temperature (what we vary in the simulations is the reduced temperature $T = \epsilon/k_B$). In soft matter systems, the experiments are often performed at room temperature and the interaction strength ϵ can be varied (by changing the solvent or ions concentration for example) although there are cases in which instead the temperature T is varied. In all cases, $k_B T/\epsilon$ is the relevant parameter and, in our interpretation, decreasing $k_B T/\epsilon$ has the role of decreasing the strength of thermal fluctuations with respect to the one of stress heterogeneities frozen-in during solidification (whose typical energy scale is of the order of ϵ).

In experiments, an increase in β (from $\beta < 1$ to $\beta > 1$) with decreasing T close to T_g and beyond has been observed for glass-forming polymers by H. Conrad et al PRE 2015 (Ref.[23] in the revised manuscript).

Furthermore, there are other indications from experiments that increasing frozen-in internal stresses may lead to a similar change in the aging dynamics: in B. Ruta et al., Soft Matter 2014 (Ref.[24]), the authors study a biopolymer physical gel (methylcellulose) and show an increase in β (from stretched to compressed) by increasing the temperature, which in their system increases the strength of the interactions (with respect to thermal fluctuations, thus effectively decreasing $k_B T/\epsilon$) and brings the system deeper in the non-ergodic gel state; a transition from stretched to compressed exponential dynamics has been observed in gelling Laponite suspensions upon increasing the Laponite volume fraction deeper in the gel (or non-ergodic) state (see F. Schosseler et al. PRE 2006 - Ref.[25]); R. Angelini et al. Soft Matter 2013 (Ref.[21]) found that the aging dynamics of a colloidal clay suspension changed from stretched $\beta < 1$ to compressed $\beta > 1$ when the initial sample had been previously subjected to shear stresses.

These last observations are also consistent with the conclusions of our work, in that they show how a different pattern in the internal stresses, and therefore differences in the spatio-temporal correlation of the stress fluctuations, may change the aging dynamics of such soft solids.

Revision: We have mentioned this point in the conclusions of the revised manuscript and added a few references.

Question : Authors cite many papers wherein compressed exponential behavior is observed (references: 18 to 25). However in most of these systems aging involves either increase in elastic modulus or that of relaxation time. While authors do plot relaxation time dependence of q , nowhere they plot relaxation

time dependence of aging time. How does it look like? Does it resemble scalings mentioned in any of the references 18-25.

ANSWER: In most of the papers mentioned by the referee, the compressed exponential behavior is associated to a slow (or ultraslow) aging regime where the relaxation time depends sublinearly (or linearly at most) on the sample age. This is in contrast to a faster aging regime (reported in some of the papers) where the relaxation time grows instead exponentially with the sample age.

Prompted by the referee, we have also run extensive additional simulations to improve the statistics for the calculation of the relaxation time (within the time constraint of this answer). In Figure 1 below, we plot the relaxation time τ_s (obtained from the incoherent scattering functions, which allow for better statistics) as a function of all values of the sample age t_w explored so far (which correspond to $\simeq 4 \times 10^7$ MD steps, requiring several weeks of computation). Overall, the relaxation time increases very slowly with the sample age.

Figure 1 (right) in these notes shows the dependence of τ_s on the waiting time t_w for one value of the wave vector $q = 1.6$ (that corresponds to length scale about 4σ and to a $\beta \simeq 1.5$). Upon increasing the sample age, eventually the relaxation time increases but the growth is only logarithmic. Such finding, in our view, indicates that the characteristic aging time of the gel is well beyond our simulation time window. In comparing with the experiments, our data seem to rule out an exponential aging regime and may be consistent with the ultra-slow, sub-linear increase (sometime referred to as *full aging*). Nevertheless, obtaining a convincing prediction for the scaling of τ_s with the sample age would require simulations that cover a significantly wider time window (i.e. 2×10^9 MD steps). Such study goes beyond the scope of this paper, but it will be part of future work.

Consistent with an ultraslow aging regime in soft gels, where the relaxation time grows slowly with the sample age, is also the observation (from the simulations) that the ruptures underlying the gel aging become extremely rare already for values of $k_B T/\epsilon$ larger than the one of interest here (roughly only a few over $\sim 10^6$ MD steps at $k_B T/\epsilon = 0.05$). It is precisely this fact that lead us to develop the approach proposed here, where we mimic microscopic ruptures underlying the aging by removing network connections associated to the largest tensile stresses.

Revision: We have added a comment about the dependence of the relaxation time on the sample age in the text at page 3.

Question : My bigger concern is that the authors express aging as rupture of interparticle bonds. However, experimentally enhancement of relaxation time/elastic modulus is observed. Doesn't it mean that with time there should be net buildup of bonds than rupture. Authors should therefore plot evolution of relaxation time/elastic modulus as a function of time for the three cases explored.

ANSWER: In our simulations the elementary aging events are indeed ruptures of the structure which will mainly tend to reduce the modulus rather than increase it. We agree with the referee that our study does not necessarily cover all possible elementary aging events that can happen in different materials (e.g. we do not consider coarsening or com-

Figure 1: Left: The structural relaxation time τ_s as a function of the wave vector s extracted from the incoherent scattering function, and for different t_w in the limit where thermal fluctuations are negligible $k_B T/\epsilon = 0$. Right: The relaxation time τ_s as a function of the sample age t_w for one typical value of the wave vector $q = 1.6$ that corresponds to length scale about 4σ , the solid blue line corresponds to a fit with $a \ln(t_w + b) + c$.

paction of initially loosely packed domains) but the approach taken is the simplest and most effective for the gel considered here. One could extend our approach to include also favoring recombinations of gel branches or local micro-compaction events (both would tend to favor an increase of the modulus over time). Such studies are highly demanding computationally and go beyond the scope of this paper, but could be the subject of future work. In all cases, we note that the disruption of the elastic strain field due to the rupture of a branch of the gel is basically the same as the one induced by a recombination of the gel branches (see Ref[29]). As a matter of fact, in our simulations, recombinations of the gel branches are actually possible but just not observed at the volume fraction and for the time window explored here.

We note that an increase in the relaxation time measured in the scattering experiments is not necessarily always coupled to an increase of the elastic modulus, since the coupling between the evolution of the viscoelastic spectrum in aging gels (or other soft materials) and the evolution of the relaxation time with the sample age is far from trivial. As a matter of fact, the elastic modulus could increase or *decrease* with the sample age, depending also (but not only) on the initial gel structure and on which type of microscopic events are driving the aging (e.g., ruptures of the gel connections vs local compaction of parts of the gel or coarsening, etc). As an example, in Lieleg et al, Nat. Mat. 2011 (Ref[41]), where an actin/fascin bundle gel network is studied, the compressed exponential dynamics and the increase of the relaxation time with the gel age are associated to a *decrease* of the modulus.

Fig.2(left) displays the viscoelastic spectrum of the gels discussed in the paper, from

Figure 2: Left: Oscillatory shear frequency sweeps performed with a strain $\gamma_0 = 1\%$. The data show the first-harmonic storage moduli G' and loss modulus G'' as a function of the frequency ω for $k_B T/\epsilon = 0$ and for different waiting times t_w . The values of t_w shown in the figure correspond respectively to the initial configuration (red), after cutting 1% of the bonds (cyan) and the final configuration after altering $\simeq 6\%$ of the interparticles bonds (orange). Right: The value of the storage modulus at the lowest frequency investigated ($\omega\tau_0 = 6.310^{-5}$) as a function of the waiting time t_w for the same values of $k_B T/\epsilon$ considered in the paper: Athermal (red), $k_B T/\epsilon = 10^{-4}$ (green) and $k_B T/\epsilon = 10^{-3}$ (blue).

which the low frequency elastic modulus G' has been extracted. G' is plotted as a function of the sample age in Fig.2(right), as asked by the referee, and shows that the modulus indeed decreases, whereas we have shown in the previous figure that the relaxation time increases (although very slowly) with the sample age.

Such findings do not represent a concern for our main findings in the paper, since the emerging dynamics (compressed or stretched) should not depend upon the specific elementary events considered but rather on the interplay between the disruption of the elastic strain field, the spatio-temporal correlations of the stresses and the thermal fluctuations.

Revision: We have included part of this discussion in the revised version of the manuscript (see text in blue page 1, 2 and page 5).

Reply to referee 2:

In this paper, the authors report on a large-scale study of a model gel through overdamped Langevin dynamics. They identify and isolate those links in the gel that carry the largest stress, and study what happens when these links are artificially removed. They find that a qualitatively different dynamics at high and low temperatures. At high temperature, the dynamics is subdiffusive with stretched exponentials, as in regular glassy materials. At low temperature, it becomes superdiffusive and shows compressed exponential decay, as in some experiments on gels that appeared in the literature recently. They attribute this difference to a change in the way elastic stresses propagate in the material. At high temperature, thermal fluctuations screen the elastic propagator leading to a Gaussian like distribution of stresses. At low temperature this screening becomes ineffective leading to a power-law distribution of stresses and a faster propagation. (This is at least my understanding of the paper). I find the result very interesting. It provides a general picture to understand the difference between glasses and gels, based on a clear microscopic phenomenon. The study will probably have an important impact on the community and lead to many follow-ups. I therefore recommend publication in Nature Communications. The results are clearly explained and the analysis looks solid and reproducible.

Question: *The introduction and conclusions to the paper could instead be improved. As such, they are quite difficult to read for a non specialist. Many technical words are used without definition (e.g. "enthalpic stresses", which by the way is a concept that is never fully defined and I did not understand the reason for this name). I would suggest a longer introduction, with a better explanation of the main concepts and a more explicit contrast between the standard glassy scenario and the new scenario proposed here.*

ANSWER: We thank the referee for her/his positive appreciation of our work.

Revision: Following the suggestion made, we have substantially revised the introduction and conclusions (see text in blue).

Reply to referee 3:

In the present manuscript (MS) the authors present a detailed numerical investigation of a specific model describing a low density gel with directional bonds. The authors investigate the dynamics during aging at different "temperature", to change the relative importance of thermal fluctuations and mechanical relaxation. Investigating the stress and the displacements associated to rupture events during the dynamics, the authors find the transition from a thermal fluctuation dominated aging at high T to a more mechanical (enthalpy) dominated aging in the athermal system. In the latter regime, the fluctuations are strongly non Gaussian. According to the authors this model well represents a class of real systems, allowing to rationalize the experimental findings. Moreover this model allows one to study the aging processes in soft matter, an issue of great importance for the practical application of these materials. I find that the work is highly professional and the MS is well written (few minor issues are listed below). Overall, I think that the MS deserves to be published in Nature Communications.

Question: There is, however, one issue that puzzle me. Concerning the intermediate scattering function, the authors find that at small q the relaxation time is q -independent (see Fig. (3), panel d). This is a strange result, not shared with other (simulated or experimental) cases (Ref. [18], cited by the authors at support of this finding, concerns fractal objects, cannot be considered a good counterpart for the present model). Intuitively the flatness of $\tau(q)$ at small q should correspond to a dynamics made of largely independent and uncorrelated (over large distance) events. How this is compatible with the authors' statement: "Such finding suggests that, although local rearrangements are possible, over larger distances the material is solid and elastically connected in spite of being structurally heterogeneous and sparse...", My feeling is that the investigated model has a dynamic basically uncorrelated over large distances. I would ask the authors to further expand this point. Indeed, the interest on the present results (i.e. its applicability to the science of materials) strongly depends on how much its dynamic is far from a collection of (spatially) isolated events.

ANSWER: We thank the referee for his/her dedicated reading of our manuscript and the positive appreciation of our work.

We also thank the referee for the opportunity to clarify this point. The dynamics resulting from the gel aging is indeed correlated over length scales of the order of the system size and for all cases explored here. Fig.4(a) of the manuscript shows that the distribution of the microscopic displacements is never Gaussian. At the highest $k_B T/\epsilon$ explored here, such distribution features exponential tails that are consistent with cooperative glassy dynamics and, upon decreasing $k_B T/\epsilon$, it displays a power law tail fully consistent with long range elasticity (and hence with the presence of long-range spatial correlations). In addition, Fig. 5(c) of the manuscript shows ultraslow correlations in the stress fluctuations, which would

not be possible without long-range spatial correlations in the relaxation dynamics.

The weak dependence of the relaxation time τ on the wave vector at low q , displayed in Fig. 3(d) of the manuscript, is specific to the data obtained from the *coherent* scattering function, which quantifies the correlations in the density fluctuations between regions of the gel that are separated by a distance determined by the wave vector. In particular, when varying the wave vector, one is detecting the relative change of correlations between points in the material separated by a certain distance. Whereas we agree with the referee that such weak dependence could be interpreted as an absence of spatial correlations, in the context described above the data in Fig. 3(d) of the manuscript can be better understood by considering that, although local rearrangements are possible, over larger distances the material is solid and elastically connected in spite of being structurally heterogeneous and sparse. In other words, regions separated by distances larger than the average mesh size are elastically connected and hence responding coherently to the strain perturbation induced by the cutting. As a consequence, density fluctuations at different points (over large distances) are always strongly coherent and the relative degree of correlations does not change much upon increasing the distance between the 2 points probed. Also in such case one would measure a relatively weak change of the relaxation time with varying the size of the region over which the dynamics is being sampled.

Such behavior should be accessible to scattering experiments, provided they can go to probe low enough q values (where low enough depends of course on the type of system investigated). Therefore the data available in the literature are not necessarily in contrast with our finding and the data in Ref. [18] (Ref.[20] in the revised manuscript) may well be consistent with it, since our explanation should hold also for fractal systems.

We note from Fig. 3(d) of the paper that the weak q -dependence of τ persists also when thermal fluctuations increase, indicating that, in the whole range of $k_B T/\epsilon$ investigated here, there is relatively little structural relaxation that can happen at the network level, even when thermal fluctuations start to help relax stress correlations. This is because the system is in all cases non-ergodic over the simulation time window and the network cannot significantly restructure over the simulation time window.

It is important to note that the q -dependence discussed so far is different from what one would obtain from the *incoherent* scattering function, which captures instead the time correlations in the single particle displacement: in this case, decreasing the wave vector corresponds to increasing the linear size of the region over which the displacements are being measured, hence entailing larger displacements and longer relaxation times. In Fig.3 in these notes (also Figure 1 of the Supplementary Information of the manuscript) we show the q -dependence for the relaxation time $\tau_s(q)$ extracted from the incoherent scattering functions in our simulations. As expected, τ_s has a stronger dependence on q at low q and in the inset of the figure one can see that in the athermal regime the scaling $\tau_s(q) \propto q^{1.35}$ holds for $q < q^*$. Upon increasing $k_B T/\epsilon$ (with the gel still being an elastic solid) the incoherent scattering data at low q display a stronger dependence and a trend toward the diffusive scaling $1/q^2$, due to transition from the elastically driven, intermittent dynamics to the glassy stretched exponential dynamics.

Finally, our explanation of the data in Fig. 3(d) of the paper is consistent with the

Figure 3: The structural relaxation time τ_s as a function of the wave vector q extracted from the incoherent scattering function for $k_B T / \epsilon = 0$. The inset shows the scaling for small wave vectors $\tau_s \propto q^{-1.35}$.

agreement found in our study (and in many experiments) with the mean field predictions of Ref. [29] for the compressed exponential dynamics, in spite of the strong spatial heterogeneities in the structure of the gels. The mean field approximation holds in this case not because of absence of correlations but rather because the correlations are always very strong and relative changes cannot be detected. This arises from the fact that the fluctuations dominating the compressed exponential regime are elastic in nature, and hence remain strongly spatially correlated (since the elasticity emerges directly from the connectivity of the structure and hence from its spatial organization).

Revision: We have revised the text to better clarify this point (see text in blue page 4).

Question: Minor issues:

1-In the displayed equation (section "Microscopic dynamics") defining $F_{t_w}(q, \tau)$, in the argument of the exponential appears $(r_k(\tau) - r_j(t_w))$. Should not it be $(r_k(t_w + \tau) - r_j(t_w))$?

2-The labelling (a, b, c) of the three panels of Figure 4 is wrongly reported in the caption.

3-Figure 5. I suppose that the color code for the three panels is the same reported in the central one. For clarity, please, report the code in the caption.

ANSWER: We thank the referee for pointing out these inconsistencies.

Revision: We have corrected all the issues indicated by the referee.

REVIEWERS' COMMENTS:

Reviewer #1 (Remarks to the Author):

Comments on the revised manuscript: Elastically driven, intermittent microscopic dynamics in soft solids, by Bouzid and coworkers.

I went through the revised version and the response to the reviewer's comments. The authors have done a careful job of carrying out the 3D simulations of colloidal gel, and the results are indeed interesting. The question is whether the work is interesting enough to merit publication in Nature Communications. In my opinion, the results that they show are yet to realize experimentally. The experimental systems that they cite in response to my previous comments are very different than what they are studying. They primarily cite two colloidal systems that they claim show similar experimental behavior: Aqueous Laponite suspension and a natural biopolymer physical gel. In the first system, in one of the papers, change from stretched to compressed is observed with age; while in the second paper the behavioral change from the stretched to compressed occurs when sample is subjected to the deformation field. It should be noted that Laponite is plate-like in shape, and therefore effect of deformation is expected to have non-trivial effect on the microstructure that is different than just increase in mobility. In the second system change-over from stretched to compressed occurs with increase in temperature, opposite of what Bouzid and coworkers report. Furthermore, Laponite suspension shows huge enhancement in G' as a function of age, the present system on the other hand shows decrease in G' .

Overall, there is practically no experimental evidence of the behavior reported by the authors. Moreover, the aging mechanism that they propose is followed by a small subset of experimental systems. Under such circumstances, I fail to appreciate broad scope of the present work that is necessary for Nature Communications. I therefore do not find the paper suitable for Nature Communications. Having said this, I do not raise any questions on the scientific merit or the rigor of the work carried out, which I think makes the paper suitable for journals like Soft Matter, Journal of Chemical Physics, or Physical Reviews E.

Reviewer #2 (Remarks to the Author):

I examined the two other referee reports, and the answers of the authors to all referees together with the changes made to the manuscript.

My overall impression is that the authors addressed correctly and carefully all the remarks.

I confirm my previous statement that the results of this paper are very important. In my opinion, the revised manuscript deserves publication on Nature Communications as it is.

Reply to Referee 1:

Comment : I went through the revised version and the response to the reviewer's comments. The authors have done a careful job of carrying out the 3D simulations of colloidal gel, and the results are indeed interesting. The question is whether the work is interesting enough to merit publication in Nature Communications. In my opinion, the results that they show are yet to realize experimentally. The experimental systems that they cite in response to my previous comments are very different than what they are studying. They primarily cite two colloidal systems that they claim show similar experimental behavior: Aqueous Laponite suspension and a natural biopolymer physical gel. In the first system, in one of the papers, change from stretched to compressed is observed with age; while in the second paper the behavioral change from the stretched to compressed occurs when sample is subjected to the deformation field. It should be noted that Laponite is plate-like in shape, and therefore effect of deformation is expected to have non-trivial effect on the microstructure that is different than just increase in mobility. In the second system change-over from stretched to compressed occurs with increase in temperature, opposite of what Bouzid and coworkers report. Furthermore, Laponite suspension shows huge enhancement in G' as a function of age, the present system on the other hand shows decrease in G' .

Overall, there is practically no experimental evidence of the behavior reported by the authors. Moreover, the aging mechanism that they propose is followed by a small subset of experimental systems. Under such circumstances, I fail to appreciate broad scope of the present work that is necessary for Nature Communications. I therefore do not find the paper suitable for Nature Communications. Having said this, I do not raise any questions on the scientific merit or the rigor of the work carried out, which I think makes the paper suitable for journals like *Soft Matter*, *Journal of Chemical Physics*, or *Physical Reviews E*.

ANSWER: As thoroughly clarified in the paper and in the previous answer to all Referees' comments, we do not claim that the way we control the presence of thermal fluctuations in the simulations (e.g., just by varying $k_B T/\epsilon$, while keeping all other microstructural details equal) corresponds to any of the experiments mentioned or cited. The true value of using numerical methods and theoretical modeling is *not* in mimicking the details of an experimental system but rather in being able to abstract and distill the fundamental physical mechanisms underlying the phenomenon of interest in ways that are not accessible to experiments. We believe we have achieved this here.

Our main finding is that in soft solids stress heterogeneities frozen-in upon solidification can partially relax through elastically driven fluctuations, when thermal fluctuations are too weak. Such stress fluctuations are intermittent in nature, because of strong correlations that persist over the timescale of experiments or simulations, leading to faster than

exponential dynamics.

With respect to the case of the Laponite gel mentioned by the Referee, we have thoroughly explained in our previous answer (and in the paper) that, according to our results, the compressed exponential is *not* the manifestation of a mere mobility increase but the consequence of the partial relaxation of frozen-in internal stresses in an elastic environment. We suggest that this is what may be happening also in the Laponite gels (in some form that certainly depends on the microscopic details of the Laponite gel microstructure and that is not the focus of our work or model used). The experiments on the Laponite gel suggest, in our view, that the emergence of the compressed exponential decay is controlled by the spatial distribution of the stresses frozen-in upon solidification, which is indeed modified by the pre-shearing protocol exactly for the reason that the Referee mention (the non-trivial changes in the microstructure). These facts are consistent with our explanation of the phenomenon. In the previous answer to the Referee’s comments, we have also provided a thorough and coherent explanation for a different dependence of the gel modulus upon aging in *different* types of materials and different aging conditions. We have clarified similarities and differences of the model used in the simulations with different experimental systems. Therefore, we fail to appreciate the relevance of the Referee’s remark.

As a matter of fact, we do not aim at mimicking specifically one of the materials mentioned by the Referee and we have never stated so. Our study focus on more general feature of soft amorphous solids (i.e., elasticity, frozen-in stresses, sensitivity to thermal fluctuations) which are shared indeed by *all* of the experimental systems in refs. [1-15], [20-29], [44], [45], and more. In fact, we are able not only to explain the emergence of the compressed exponential dynamics, but also to identify the conditions in which such dynamics would be prevalent or not, in different materials. As a consequence, our work does not concern “a small subset of experimental systems” and has indeed a broad scope.